# Longwave Radiative Effect of the Cloud-Aerosol Transition Zone Based on CERES Observations

**Babak Jahani[1], Hendrik Andersen[2,3], Josep Calbó[1], Josep-Abel González[1], Jan Cermak[2,3]**

[1] Departament de Física, Universitat de Girona, Girona, Spain

[2] Institute of Meteorology and Climate Research, Karlsruhe Institute of Technology (KIT), Karlsruhe, Germany

[3] Institute of Photogrammetry and Remote Sensing, Karlsruhe Institute of Technology (KIT), Karlsruhe, Germany

*Correspondence to*: B. Jahani (babak.jahani@udg.edu)

**Abstract.** This study presents an approach for quantification of cloud-aerosol transition zone broadband longwave radiative effects at the top of the atmosphere (TOA) during daytime over the ocean, based on satellite observations and radiative transfer

simulation. Specifically, we used several products from MODIS (Moderate Resolution Imaging Spectroradiometer) and CERES (Clouds and the Earth's Radiant Energy System) sensors for identification and selection of CERES footprints with horizontally homogeneous transition zone and clear-sky conditions. For the selected transition zone footprints, radiative effect was calculated as the difference between the instantaneous CERES TOA upwelling broadband longwave radiance observations and corresponding clear-sky radiance simulations. The clear-sky radiances were simulated using the Santa Barbara DISORT

Atmospheric Radiative Transfer model fed by the hourly ERA5 reanalysis (fifth generation ECMWF reanalysis) atmospheric and surface data. The CERES radiance observations corresponding to the clear-sky footprints detected were also used for validating the simulated clear-sky radiances. We tested this approach using the radiative measurements made by the MODIS and CERES instruments onboard the Aqua platform over the south-eastern Atlantic Ocean during August 2010. For the studied period and domain, transition zone radiative effect (given in flux units) is on average equal to $8.0 \pm 3.7$ W m$^{-2}$ (heating effect;

median: 5.4 W m$^{-2}$), although cases with radiative effects as large as 50 W m$^{-2}$ were found.

## 1 Introduction

Cloud and aerosol are the particular names for two specific particle suspensions in the atmosphere, which have been widely studied but continue to contribute the largest uncertainty to estimates and interpretations of the Earth's changing energy budget (Boucher et al., 2013). One of the sources of this uncertainty is the fact that they are univocally differentiated in the atmospheric science, whereas clouds and aerosols co-exist and often interact with each other, making it hard to study the effects of one without considering the other. For instance, aerosols in the vicinity of clouds are usually hydrated in part, and their size distribution and thus their optical characteristics change relative to their dry counterpart (Várnai & Marshak, 2011). On the other hand, aerosols also affect cloud optical and microphysical properties through acting as cloud condensation nuclei and ice nucleating particles (Rosenfeld et al., 2014). Moreover, the decision on what a cloud is, or in other words where the boundaries of the clouds should be put, is a point of debate (Bar-Or et al., 2011; Fuchs and Cermak, 2015; Calbó et al., 2017; Eytan et al., 2020) and a suspension detected as cloud by one method may be labeled differently by another. This is due to the presence of special conditions called the transition zone (or twilight zone) in the region between the cloudy and so-called cloud-free skies, at which the characteristics of the suspension lay between those corresponding to the adjacent clouds and the surrounding aerosol (Koren et al., 2007; Várnai et al., 2013). These conditions consist of a mixture of liquid droplets and humidified to dry aerosols, and involves various processes such as cloud dissipation/formation, aerosol hydration/dehydration, shearing of cloud fragments, clouds becoming undetectable, etc. (Eytan et al., 2020; Koren et al., 2009).

Observations have shown that the transition zone occurs often over large areas. According to Koren et al. (2007), at any time almost 30-60% of the global atmosphere categorized as clear sky (cloud-free) can potentially correspond to these conditions, which may expand up to 30 kilometers away from the detectable clouds (Bar-Or et al., 2011). On the basis of three ground-based observation systems, Calbó et al. (2017) quantified, at two mid-latitude sites, that the frequency of the transition zone was about 10%. A global analysis based on MODIS (Moderate Resolution Imaging Spectroradiometer) products performed by Schwarz et al. (2017) also suggests a frequency of 20% for the occurrence of the transition zone.

If the area covered with the suspension of particles with the characteristics of the transition zone is so vast, the question "what role does the transition zone play in the determination of the Earth's energy budget?" takes a great importance. However, as the information available about the transition zone and its interactions with radiation (in both longwave and shortwave bands) is very limited, the area corresponding to the transition zone in climatic, meteorological, and atmospheric studies and models is usually considered as an area containing either aerosols or optically thin clouds. This means that either radiative properties of clouds or those of aerosols are used to describe the radiative properties of the transition zone. Based upon sensitivity analysis performed using radiative transfer parameterizations, two recent studies (Jahani et al. 2019, 2020) showed that this assumption may lead to substantial differences in the simulated broadband shortwave and longwave radiative effects. According to these studies, for some particular situations, at an optical depth of 0.1 (at 0.550 μm) the differences at surface and top of the

atmosphere may be as large as 7.5 and 28 W m$^{-2}$ in broadband longwave and total shortwave, respectively. Based upon an observational and statistical study, Eytan et al. (2020) estimated the top of the atmosphere (TOA) radiative effect of the transition zone around shallow warm clouds in the atmospheric window region (8.4-12.2 μm). They found that over the oceans on average the transition zone radiative effect in the mentioned spectral region is about 0.75 W m$^{-2}$ (although they found cases with average radiative effects as large as 4 W m$^{-2}$), which is equal to the radiative forcing resulting from increasing atmospheric $CO_2$ by 75 ppm. The overall radiative effects of the transition zone are likely to be higher, as the radiative effect estimations given in the latter study correspond to a lower bound of the effect and are limited to the low-level (warm) transition zone conditions. These results highlight the importance of the characterization of the transition zone as well as of quantifying the role it plays in the determination of Earth's energy budget.

Although the transition zone is frequently neglected in cloud-aerosol related studies, the above numbers and the vast area that potentially may contain the transition zone state give importance to the necessity of further exploring it. For this reason, within the frame of the study, a method for the quantification of the broadband longwave radiative effects of the transition zone at TOA over the ocean on the basis of instantaneous satellite observations and radiative transfer calculations is presented. This method is then applied over the South-Eastern Atlantic Ocean, where cloudy conditions are frequent and hence transition zone conditions are also expected to be frequently observed.

## 2 Methods

### 2.1 Satellite Observations

The CERES (Clouds and the Earth's Radiant Energy System) sensor is a three-channel scanning radiometer measuring the broadband outgoing shortwave (0.3-5 μm), window-region (8-12 μm) and longwave (5-100 μm) radiances at TOA with a spatial resolution of ~20 km at nadir regardless of the sky condition (Loeb et al., 2001; Priestley et al., 2011). The measured radiances (shortwave, window region and longwave) may then be transformed into irradiances (fluxes) according to the meteorological, physical and optical characteristics of the scene (such as suspension fraction, optical depth and phase, wind speed, etc.), using the empirical Angular Distribution Models, explained in Loeb et al. (2005). The Level-2 Single Scanner Footprint (SSF) product of the CERES instrument this instrument provides information about the measured instantaneous outgoing broadband longwave radiances at TOA as well as the corresponding estimated irradiances (Loeb et al., 2018; Loeb et al., 2006). From the SSF Level-2 product, we obtained the entire daytime instantaneous TOA outgoing broadband longwave radiance observations of the CERES instrument onboard Aqua spacecraft ($\uparrow L_{CERES}$) along with the corresponding time, geolocation, viewing geometry and surface emissivity parameters for August 2010 for the region comprised within 21$^o$ W-21$^o$ E and 10$^o$ N-50$^o$ S. We have chosen to use radiances rather than irradiances in this study to be able to provide a more direct comparison between the simulations and observations regardless of the sky condition.

In addition, several products from the MODIS (Moderate Resolution Imaging Spectroradiometer) instrument onboard the same satellite (Aqua) were used for the identification of horizontally homogeneous clear-sky and transition zone conditions within

CERES footprints. Specifically, we used the ocean products: (1) geolocation (MYD03, MODIS Characterization Support Team (MCST), 2017); (2) Aerosol-Cloud-Mask and Aerosol Optical Depth (AOD) taken from the Level-2 Aerosol (MYD04_L2, Levy et al., 2015); (3) Cloud Optical Depth (COD) from the Level-2 Cloud (MYD06, Platnick et al., 2015); and (4) Cloud Mask (MYD35, Ackerman & Frey, 2015). These products were obtained for all Aqua MODIS granules holding the following two conditions: i) corresponding to August 2010, and ii) containing at least one data point (pixel) within the region

0° E-15° E and 10° S-30° S. It turned out that the spatial extent of the data corresponding to the granules keeping these conditions covered an area between 21° W-21° E and 10° N-50° S. By combining these products, MODIS pixels were classified into the classes "Difficult", "Cloud", "Aerosol", "Clear", "Lost A", "Lost B", "Lost C" at 1-km resolution (at nadir) following the procedure explained in Schwarz et al. (2017). Among them, the pixels labeled as "Lost" are assumed to correspond to the transition zone conditions. Indeed, for these pixels neither aerosol nor cloud optical property (specifically, the variable

"Cloud_Optical_Thickness") retrievals exist, yet they are classified as containing a cloud (Lost A), a non-cloud obstruction (Lost B), or were not processed at all in the cloud masking (Lost C).

The processed MODIS data was then integrated from 1-km resolution to CERES native resolution to determine the fraction of each class and the average values of COD and AOD in the CERES footprints, considering equal weights for all MODIS pixels (the procedure adopted for matching the MODIS pixels with CERES footprints is explained in Appendix A). Afterwards, only

CERES footprints meeting all the following conditions were used in the analysis: (i) solar zenith angles and CERES viewing zenith angles at surface lower than 60° (to mitigate the effect of uncertainties derived from viewing and solar geometries), (ii) no land MODIS pixels as determined using the MYD35 data is included, and (iii) the number of MODIS ocean pixels equals or exceeds 75% of the ≈400 pixels expected to fall within the CERES field of view (FOV; to exclude FOVs located on the edges of the MODIS granules). Among the remaining footprints, those with a "Lost" fraction (all lost classes together) greater

than or equal to 90% were classified as horizontally homogeneous transition zone footprints (the transition zone footprints selected this way, may contain up to 10% of cloud contamination). Also, those having AOD and COD equal to zero, "Lost" fraction less than 10%, and "Difficult" fraction less than 10% were classified as horizontally homogeneous clear-sky footprints. Based on this classification criterion, a total number of 5441 clear-sky and 3783 transition zone footprints were detected over the South-Eastern Atlantic Ocean in August 2010. The spatial distribution of these footprints is presented in Figure 1.

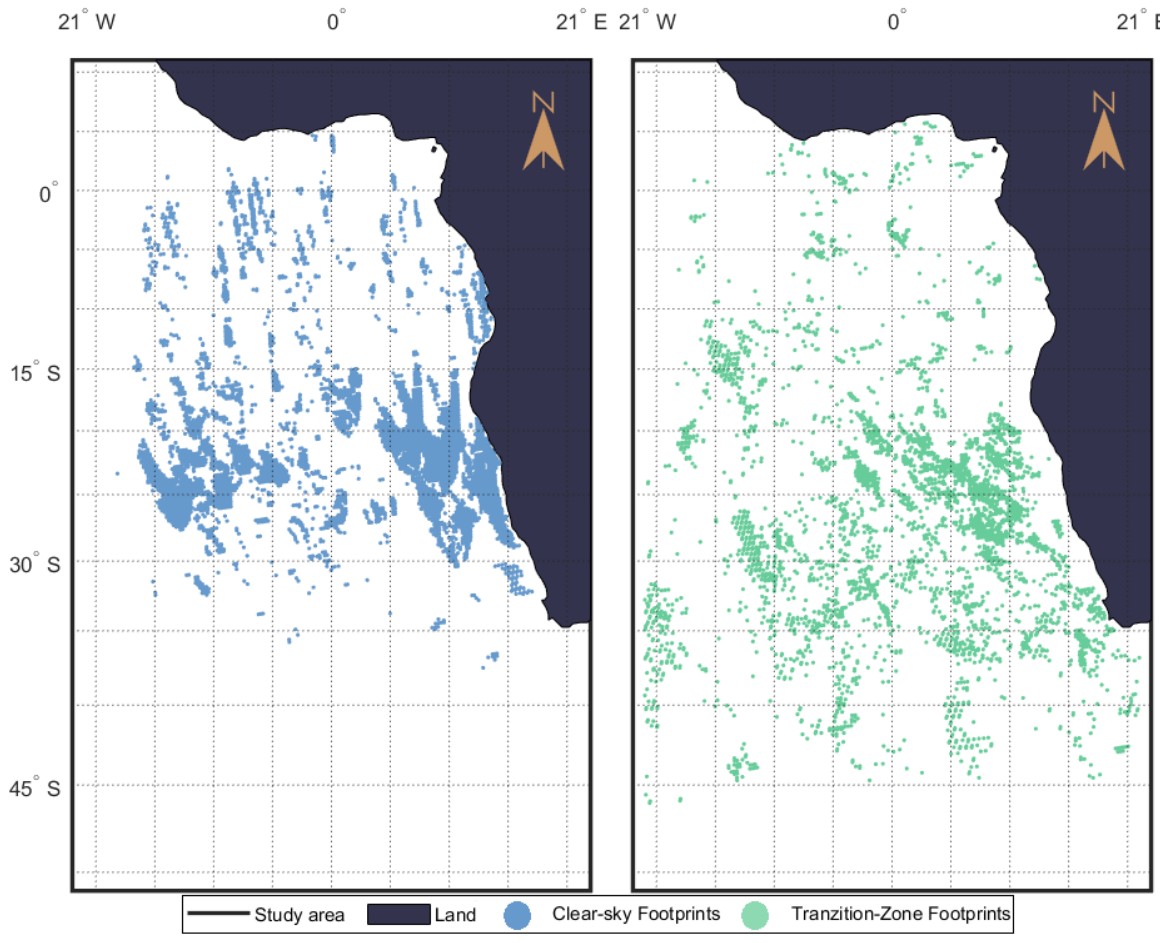


**Figure 1. Spatial distribution of the clear-sky and transition zone CERES footprints detected within the study area (21º W-21º E, 10º N-50º S) in August 2010.**

### 2.2 Clear-sky simulations

For all transition zone and clear-sky footprints selected according to the criteria explained in section 2.1, the TOA upwelling broadband longwave (5-100 μm) clear-sky radiances ($\uparrow L_{RTM,clr}$) for the CERES viewing zenith angle ($\theta$) were simulated using the Santa Barbara DISORT Atmospheric Radiative Transfer model (SBDART, Ricchiazzi et al., 1998), considering the effect of all atmospheric gases. The simulations were carried out by using atmospheric profiles (Hersbach et al., 2018a) and surface (Hersbach et al., 2018b) values provided by the fifth generation ECMWF reanalysis (ERA5), which render the data at 0.25º×0.25º spatial resolution and 1h time intervals. Specifically, profiles of specific humidity, geopotential height, ozone mass mixing ratio, and temperature at all available pressure levels (1000 hPa-1 hPa), as well as mean sea level pressure and 2 m air temperature and dewpoint temperature were used. For each (clear-sky/transition-zone) footprint, the surface and atmospheric data of the closest ERA5 cell were combined with each other and linearly interpolated in time according to the CERES time

of observation. The combined and interpolated profiles were then fed to SBDART for simulation of $\uparrow L_{RTM,clr}$. In these simulations, the broadband sea surface emissivity and the $CO_2$ concentration in atmosphere were set to the constant values of

0.982 (equal to the estimated broadband longwave sea surface emissivity included in the CERES SSF data; Geier et al., 2003) and 388.71 ppm (which is the value corresponding to the year 2010; European Environmental Agency: https://www.eea.europa.eu/, last access: 13 May 2021), respectively. As for the other gases the default concentration values included in SBDART model were used. For each individual clear-sky and transition zone footprint, SBDART model was ran with 20 zenithal streams and the spectral upwelling radiances (including the solar contribution, which actually is very low)

were calculated in the range of 5-100 μm in steps of 0.2 μm. Then, the upwelling radiances at 30 km altitude at the SBDART computational zenithal angles were outputted and linearly interpolated to determine the magnitude of the upwelling radiance in the direction $\theta$. Throughout this paper, we give negative sign to the physically upwelling radiances.

The simulated clear-sky radiances ($\uparrow L_{RTM,clr}$) were then validated through comparing them with the $\uparrow L_{CERES}$ values corresponding to the clear-sky footprints ($\uparrow L_{CERES,clr}$). The comparison was made using the corresponding isotropic irradiances

($\pi\uparrow L_{CERES,clr}$ and $\pi\uparrow L_{RTM,clr}$), and was based on the linear correlation coefficient between the simulated and the measured values, as well as by analyzing the probability distribution, mean and variance of the differences. First, for each individual clear-sky footprint, the difference between the calculated and observed clear-sky upward irradiances ($\varepsilon_{clr}$, W m$^{-2}$) was determined according to Eq. 1:

$$\varepsilon_{clr} = \pi\uparrow L_{RTM,clr} - \pi\uparrow L_{CERES,clr} \hspace{4cm} \text{Eq. 1}$$

Second, outliers were removed from the dataset by applying the quartiles method. Thus, among all clear-sky footprints (5441 footprints), those with a $\varepsilon_{clr}$ more than 1.5 interquartile ranges above the upper quartile or below the lower quartile (197 footprints) were discarded. Statistical analysis of the $\varepsilon_{clr}$ values corresponding to the remaining clear-sky footprints showed that $\pi\uparrow L_{RTM,clr}$ and $\pi\uparrow L_{CERES,clr}$ values are strongly correlated ($r^2 = 0.96$) and that $\varepsilon_{clr}$ values are normally distributed around the mean value (hereafter denoted as $\bar{\varepsilon}_{clr}$) of about 8.0 W m$^{-2}$ with a standard deviation of about 1.9 W m$^{-2}$. The probability

distribution of the $\varepsilon_{clr}$ values around $\bar{\varepsilon}_{clr}$ and the scatter plot of $\pi\uparrow L_{CERES,clr}$ versus $\pi\uparrow L_{RTM,clr}$ values shifted by $\bar{\varepsilon}_{clr}$ are provided in Figure 2.

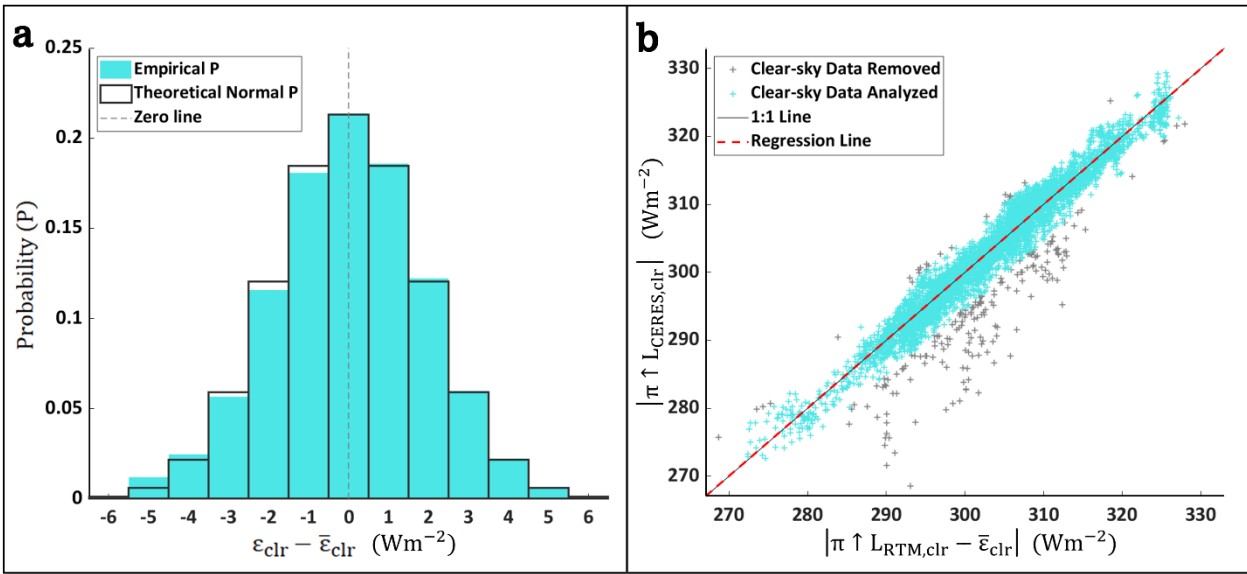

**Figure 2. (a)** Empirical (solid turquoise fill) and fitted theoretical normal (black lines) probability distributions of $\varepsilon_{clr}$ around $\bar{\varepsilon}_{clr}$, **(b)** scatter plots of $\pi{\uparrow}L_{CERES,clr}$ versus $\pi{\uparrow}L_{RTM,clr}$ values shifted by $\bar{\varepsilon}_{clr}$ in absolute sense. In Figure 2b, the gray points show the (outlier) data points discarded based on the quartile method.

The facts that $\varepsilon_{clr}$ values are normally distributed around $\bar{\varepsilon}_{clr}$ and that clear-sky observations and simulations are strongly correlated (and with a slope of the linear fit very close to 1) confirm that $\pi{\uparrow}L_{RTM,clr}$ values are systematically biased by about 8.0 W m$^{-2}$ ($\bar{\varepsilon}_{clr}$) in comparison with the $\pi{\uparrow}L_{CERES,clr}$ values. As the upwelling irradiances/radiances are negative by definition, the bias found indicates an underestimation of the simulation in absolute terms. Also, the distribution of $\varepsilon_{clr}$ values shows that a random disagreement of about ±3.7 W m$^{-2}$ at 95% confidence level (two-tailed, $\varepsilon_{clr,95}$) exists between the clear-sky observations and simulations. The bias and the random disagreement must be due to the combined effect of the uncertainties associated with the data utilized and the assumptions made in the radiative transfer simulations (such as plane parallel atmosphere assumption, number of streams used in the calculations), the spectral resolution at which the radiative transfer calculations were performed (SBDART is based on LOWTRAN band models, and it was found by Wacker et al., (2009) that the spectrally integrated clear-sky downwelling longwave irradiances simulated by LOWTRAN models is systematically 6 Wm$^{-2}$ lower in comparison with line-by-line models or using the high resolution MODTRAN correlated-k bands), temporal and spatial matching of the ERA5 profiles with the CERES footprints, and the uncertainties associated with measuring the $\uparrow L_{CERES}$. According to Loeb et al. (2001), up to 0.2% of error with a standard deviation of 0.1% is associated with $\uparrow L_{CERES}$ which is indeed measured by subtracting the radiances received at the shortwave and total channels of the sensor with the appropriate spectral correction coefficients (not directly measured). Some proportion of this error could also be due to the longwave radiation scattered from the adjacent CERES footprints, which should be rather small as the magnitude of scattering by atmospheric particles for the wavelengths between 5 and 100 μm is rather neglectable. We tested the sensitivity of $\varepsilon_{clr}$ values

to the input surface temperature, water vapor mixing ratio profile, surface emissivity parameters, as well as the number of zenithal streams used in the radiative transfer calculations. We found that, as expected, $\varepsilon_{clr}$ values vary considerably with very small changes in surface temperature (increasing/decreasing surface temperature by 1K will increase/reduce $\bar{\varepsilon}_{clr}$ by about 60%), whereas the effect of other parameters is very small. Given the fact that the temperature may notably vary in the first 2 meters of the atmosphere, the possible bias and uncertainties associated with the ERA5 surface data utilized could possibly explain some parts of the disagreements (bias and uncertainty) observed between $\pi{\uparrow}L_{CERES,clr}$ and $\pi{\uparrow}L_{RTM,clr}$.

## 2.3 Transition Zone Radiative Effects

The broadband longwave (5-100 μm) radiative effect on flux (assuming an isotropic distribution for the radiance) for the transition zone footprints ($RE_{trz}$, W m$^{-2}$) was calculated as the difference between the radiances measured by CERES ($\uparrow L_{CERES,trz}$) and the corresponding simulated clear-sky values ($\uparrow L_{RTM,clr}$) according to Eq. 2:

$$RE_{trz} = \pi \uparrow L_{CERES,trz} - \left( \pi \uparrow L_{RTM,clr} - \bar{\varepsilon}_{clr} \right) \qquad \text{Eq. 2}$$

In this equation, $\bar{\varepsilon}_{clr}$ is included for canceling the systematic bias in the estimation of $\uparrow L_{RTM,clr}$ (see Section 2.2). According to the uncertainty assessment described in section 2.2, a random error of about $\pm 3.7$ W m$^{-2}$ (at 95% confidence level) is associated with the $RE_{trz}$ values calculated this way. Worth mentioning that as in the present study we have given negative sign to the physically upwelling radiances, a positive and negative $RE_{trz}$ will imply heating and cooling effects, respectively. Also, it should be noted that $RE_{trz}$ values determined this way are indeed RE on radiance, despite they are presented in irradiance units (W m$^{-2}$) assuming isotropic radiance.

## 3 Results and Discussion

Figure 3 shows the probability distribution of the $RE_{trz}$ values obtained from analyzing the 3783 transition zone CERES footprints detected over the South-East Atlantic region during August 2010 based on the criteria and methods explained in section 2. In this figure, the left and right axis show the cumulative and absolute empirical probabilities of $RE_{trz}$, respectively.

The bar chart shows the mean fraction of the three MODIS lost classes (A, B and C) along with the fraction of "other" classes (or "non-lost" classes, i.e.: "Cloud", "Aerosol", "Difficult" and "Clear") combined in the CERES transition zone footprints analyzed. From this figure it can be seen that although the criteria explained in section 2.1 for the selection of transition zone footprints allows up to 10% of contamination by "non-lost" classes, the fraction of these classes combined in the transition zone footprints analyzed is on average about 5%. Furthermore, this figure shows that "Lost A" is the most frequent class

among all the "Lost" classes, followed by "Lost B" and "Lost C", which is in line with the results of Schwarz et al. (2017). The absolute probability of the $RE_{trz}$ values provided in Figure 3 shows that for the studied period and domain the $RE_{trz}$ values extend from -4 to 50 W m$^{-2}$ and follow a right-skewed distribution with a mean and median of about 8.0 and 5.4 W m$^{-2}$, respectively. Among these values, a vast majority (84%) of them are positive. This implies that, as expected, for the vast majority of the transition zone CERES footprints analyzed, the outgoing longwave radiation at TOA was smaller than what it

would have been if no suspension was present (as in the present study the upwelling radiances have been indicated with negative signs). In other words, the results show that at most of these footprints, a suspension of particles exists which cannot be classified as cloud or aerosol, but it is clearly interacting with the longwave radiation emitted from the sea surface and causing a reduction in the outgoing longwave radiation at TOA (heating effect). The information provided in Figure 3 also shows that for around 60% of the cases analyzed, the magnitude of the interactions of this suspension is indeed greater than

that of the uncertainties associated with the methodology adopted ($\pm 3.7$ W m$^{-2}$). These facts prove that the radiative effects shown in Figure 3 are not coincidental; contrarily, they must be due to the transition zone particle suspension. They also prove that the transition zone occurs over a vast area which makes it possible to observe its TOA radiative signature in radiative measurements at a spatial resolution as coarse as that of CERES. The heating effects corresponding to the transition zone footprints with the magnitude of $RE_{trz}$ greater than that of the method uncertainty must be due to the absorption of the longwave

radiation emitted from the sea surface by the transition zone particles and the subsequent emission by the same particles at a temperature which is considerably cooler than that of the sea surface.

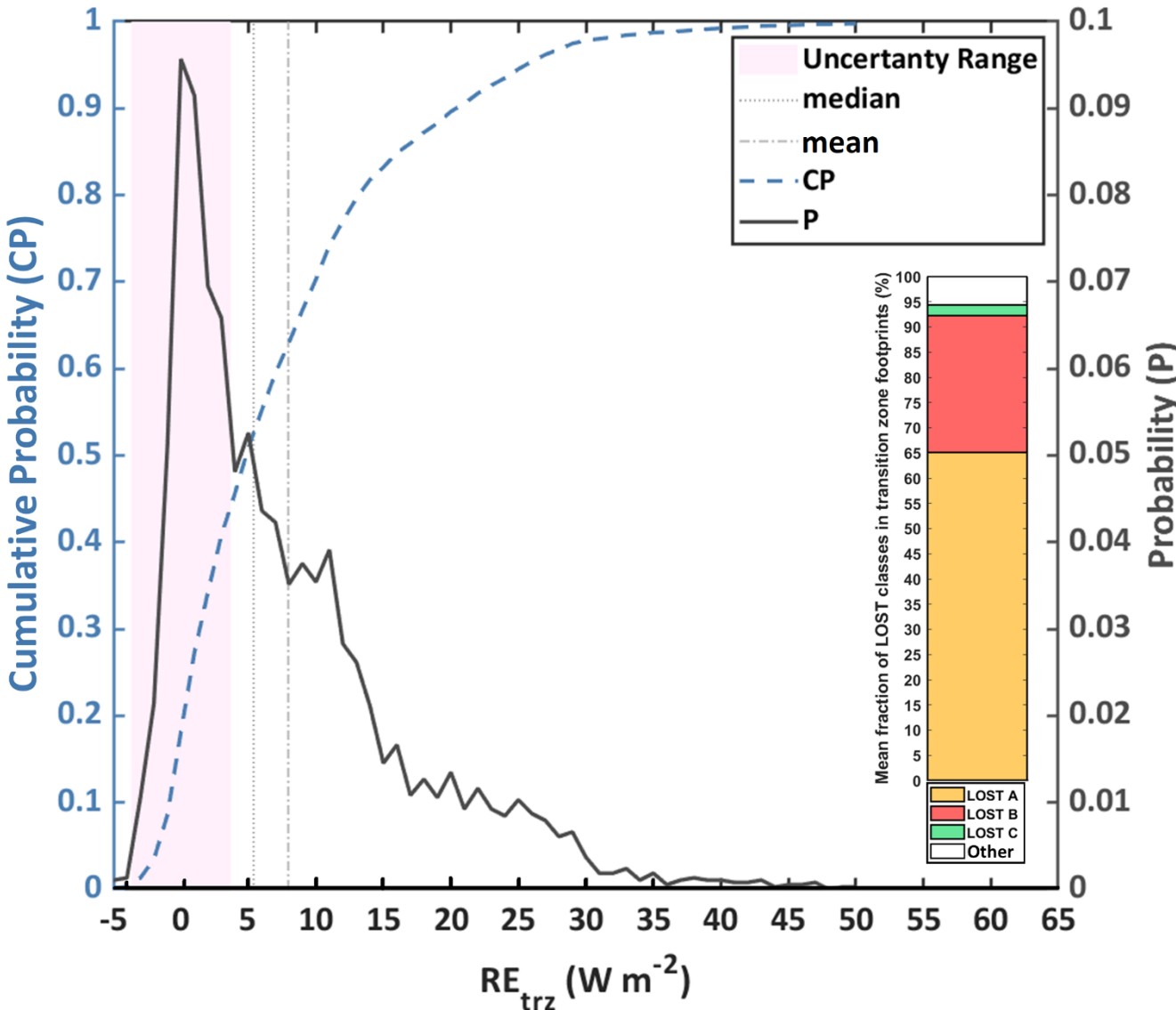

**Figure 3. Empirical cumulative (left axis) and absolute (right axis) probability distributions of the $RE_{trz}$ calculated for the 3783 transition zone footprints selected in the South-East Atlantic Ocean during August 2010. In this figure, the $RE_{trz}$ bins are 1 W m$^{-2}$ wide and centered at each enter number. The area colored in pink shows the uncertainty range, which was obtained through validating the $\pi\uparrow L_{RTM,clr}$ against $\pi\uparrow L_{CERES,clr}$ (for more information refer to Figure 2-a). The bar chart shows the mean fraction of the "Lost" classes in the transition zone footprints analyzed. The white area named as "other" in the bar chart represents the mean fraction of the classes "Difficult", "Cloud", "Aerosol" and "Clear" combined. (Note: sum of all fractions given in the bar chart equals 100%).**

The fact that the probability distribution of the $RE_{trz}$ values is right-skewed and has a tail extending up to 50 W m$^{-2}$, also indicates that the REs calculated in the present study are due to particle suspensions between cloud-free and cloudy skies with

different micro- and macro-physical characteristics, which is to be expected and is consistent with what is referred to as the transition zone: a special condition in the region between the cloudy and so-called cloud-free skies, at which the characteristics of the suspension lay between those corresponding to the adjacent clouds and the surrounding aerosols (Koren et al., 2007; Várnai et al., 2013). Among all $RE_{trz}$ values illustrated in Figure 3, for example, almost 41% of them are within the uncertainty range (-3.7 ≤ $RE_{trz}$ ≤ 3.7 W m$^{-2}$). These REs, which comprise almost all (98%) negative and 30% of the positive $RE_{trz}$ values, could potentially represent transition zone conditions with characteristics very close to clear-sky condition (relatively low concentration of particles), or those at which the upward emission by transition zone suspension is performed at temperatures close to the sea surface temperature.

Nevertheless, it should be noted that as the criteria considered for selection of horizontally homogeneous transition zone footprints allows up to 10% of contamination by other atmospheric suspensions (see section 2.1), the $RE_{trz}$ values given in Figure 3 could be partly affected by the REs of cloud edges and aerosols present in the subpixel scale. For example, the extreme values at the right tail of the distribution ($RE_{trz}$ values greater than 27.6 W m$^{-2}$, that is the 3.7% highest values) correspond to transition zone footprints contaminated with the edges of optically thick clouds. Furthermore, although we assume that the MODIS pixels classified as "Lost" correspond to the transition zone conditions, we cannot be totally sure that this assumption is true for all of the lost cases analyzed in our study. That is because there may exist lost pixels with fully developed scattered clouds present at subpixel scale. To quantify the influence of subpixel clouds on the calculated REs, we analyzed the magnitude of $RE_{trz}$ values given in Figure 3 as a function of cloud fraction in the selected CERES transition zone footprints. To perform this analysis, we clustered the transition zone CERES footprints based on the calculated cloud fraction into 10 cloud fraction bins ranging between 0 and 10 % (each bin is 1% wide). Then, for each individual cloud fraction bin, we calculated the bootstrap mean. Specifically, we selected 1000 sample groups (population of each group: 50; random sampling with replacement) from each individual cloud fraction bin and calculated the mean RE for each sample group. Afterwards, the overall average RE (and the corresponding standard deviation) for each cloud fraction bin was calculated from the sample group means. The results obtained from this analysis are presented in Figure 4. From this figure, it can be seen that $RE_{trz}$ increases with cloud fraction, which confirms the abovementioned statement about the effect of clouds on some of the calculated $RE_{trz}$ values. Nevertheless, this figure also shows that for more than 75% of the CERES transition zone footprints that we have analyzed, the cloud fraction is below 5%. While this implies that the overall effect of subpixel clouds on the calculated $RE_{trz}$ is likely to be small, we conclude that our estimation of the $RE_{trz}$ should best be interpreted as an upper bound of the longwave radiative effects of the transition zone.

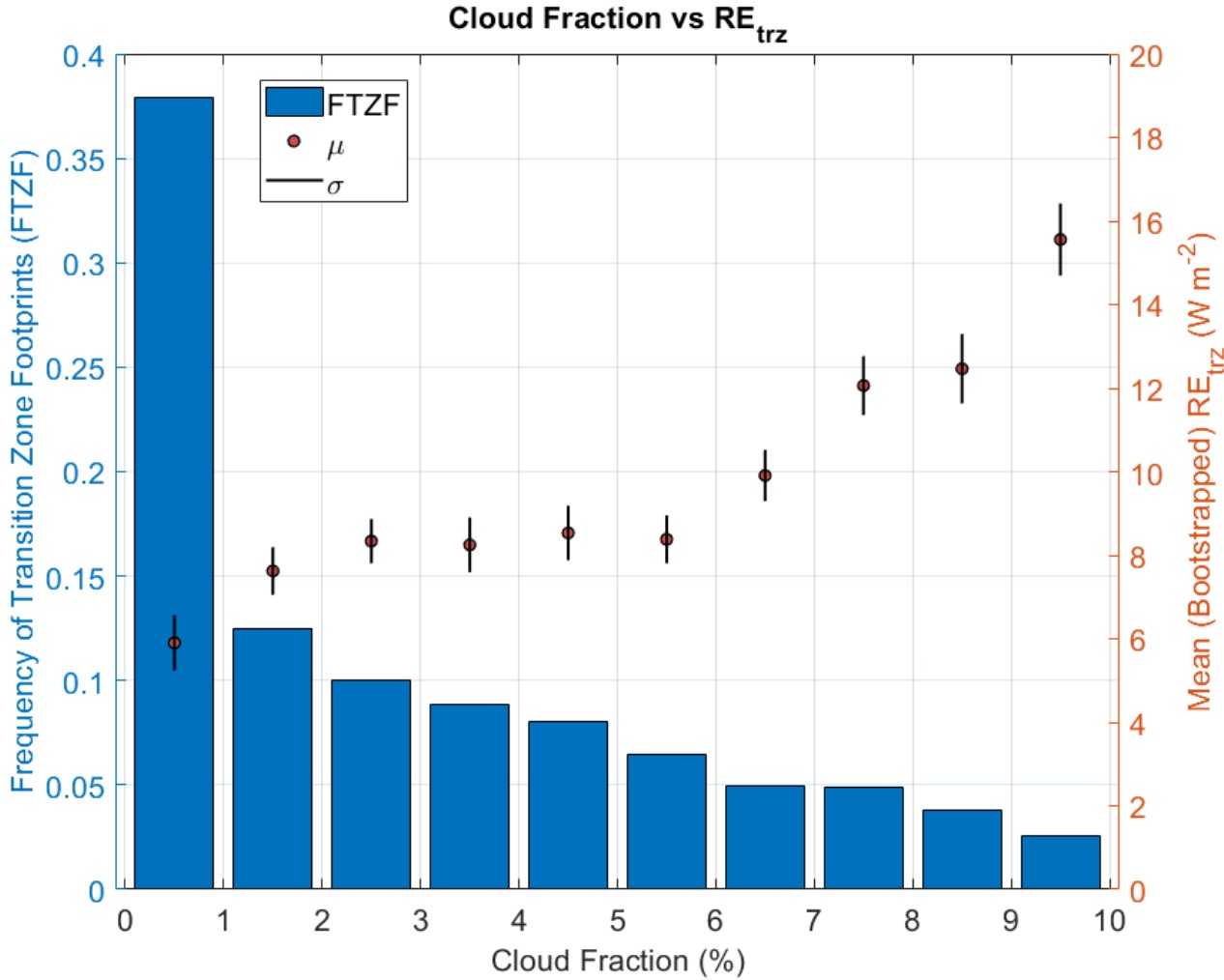

**Figure 4. RE_trz as a function of cloud fraction in the selected CERES transition zone footprints. Each cloud fraction bin given in this figure is 1% wide and the bar charts indicate the frequency of the transition zone footprints falling within the limits of each cloud fraction bin (note that the cloud fraction cannot exceed 10% because we have allowed up to 10% of non-lost contribution in the transition zone footprints selected). The red circles and black vertical lines indicate the (bootstrapped) mean RE_trz (μ) and the corresponding standard deviation (σ) for each cloud fraction bin, respectively.**

The difference between the temperatures at which the emission is performed (dT, K), specifically between the sea surface and the top of a parcel of an atmospheric particle suspension, plays a primary role in the longwave RE of this suspension at TOA. It also provides some descriptive information about the characteristics of the particle suspension. To be able to further characterize the transition zone conditions detected within the study area, dT was approximated for each transition zone footprint as the difference between the near surface air temperature and the suspension top temperature if the MODIS pixels labeled as "Lost A" were clouds. In this approximation sea surface temperature was taken equal to the ERA5 reanalysis 2 m

air temperature corresponding to the closest ERA5 grid cell linearly interpolated in time according to the time of observation (i.e., the temperature used in the clear-sky simulations). Transition zone suspension top temperature was assumed equal to CERES SSF levels 2 instantaneous cloud top temperature (Minnis et al., 2011). This assumption was made because this parameter is indeed the average of MODIS cloud top temperature retrievals made for the cloudy MODIS pixels falling within CERES FOV (determined by the MODIS cloud mask). It should be noted that in case of the transition zone footprints, according to the bar chart provided in Figure 3, 65% of the MODIS pixels were labeled as "Lost A", and that for "Lost A" pixels, cloud top temperature was retrieved, as they were initially labeled as cloud by the cloud mask. In other words, for the transition zone footprints, the temperature of the top of the suspension is the result of averaging cloud top temperature retrieved for both the "Lost A" pixels and the potential cloudy pixels falling within the remaining $\leq 10\%$ of the FOV (see section 2.1 for more information).

Figure 5 shows the values of $RE_{trz}$ as a function of dT. In this figure, gray filled circular markers, yellow crosses and vertical blue lines show the mean and median and standard deviation of the $RE_{trz}$ values corresponding to each dT bin, respectively. The horizontal black lines also indicate the width of each dT bin. Furthermore, the horizontal axis given at the top of this figure associates dT with altitude (km) according to the estimates of the mean tropospheric temperature lapse rate for the study area (6.1 K km$^{-1}$) given in Mokhov and Akperov (2006). The information provided in this figure shows that $RE_{trz}$ is strongly correlated with dT and it increases with dT ($RE_{trz}$ increases with altitude), which confirms the abovementioned statement regarding the relationship between RE and temperature at which the LW radiation is emitted. From this figure it can also be seen that dT for the transition zone footprints analyzed in the present study varies between -1.5 and 31 K. This implies that the transition zone footprints selected and analyzed in the present study in fact represent transition zone conditions at different altitudes (i.e., dT increases with altitude) and with different characteristics. The transition zone footprints with relatively small dT values, for example, represent transition zone conditions near the sea surface with characteristics close to those of the low clouds. Whereas the relatively large dT values given in Figure 5 correspond to the transition zone conditions occurring at higher altitudes (reaching altitudes as high as 5 km above the mean sea level). Given this fact, the information provided in Figure 5 suggests that the majority (about 85%) of the transition zone conditions that we have studied are below 2 km (low-level clouds) and they produce on average a RE of about 4.6 W m$^{-2}$. This was indeed to be expected, as according to Adebiyi et al. (2020), low-level clouds dominate the southeast Atlantic between July and October (although mid-level clouds are as well relatively common over this region with cloud-top heights typically placed between 5 and 7 km).

Among all the transition zone footprints analyzed in this study, those falling within the first four dT bins shown in Figure 5 could match the transition zone conditions studied by Eytan et al. (2020). Indeed, the dT values corresponding to these bins cover a dT range similar to that can be derived from Figure 2 given in Eytan et al. (2020; i.e. dT < 3-4 K). $RE_{trz}$ corresponding to these footprints, which comprise around 85% of the footprints that their $RE_{trz}$ falls within the uncertainty range (see Figure 3), is on average about 0.8 W m$^{-2}$. This number is closely in agreement with what was found by Eytan et al. (2020) as the globally averaged magnitude of $RE_{trz}$ around the warm low cloud fields (~0.75 W m$^{-2}$), even though the method adopted by

them for the selection of the transition zone conditions as well as for quantifying their REs is quite different compared to what is proposed in the present study. Specifically, in Eytan et al. (2020) distance from the nearest cloud (Koren et al., 2007) was used as a statistical measure for the likelihood of finding twilight conditions and $RE_{trz}$ was calculated based on mean TOA MODIS radiance observations. In contrast, the methodology proposed in the present study is based on instantaneous CERES observations and radiative transfer calculations and is performed on the CERES spatial scale. Furthermore, our study covers only a limited area, and their study covers the global oceans.

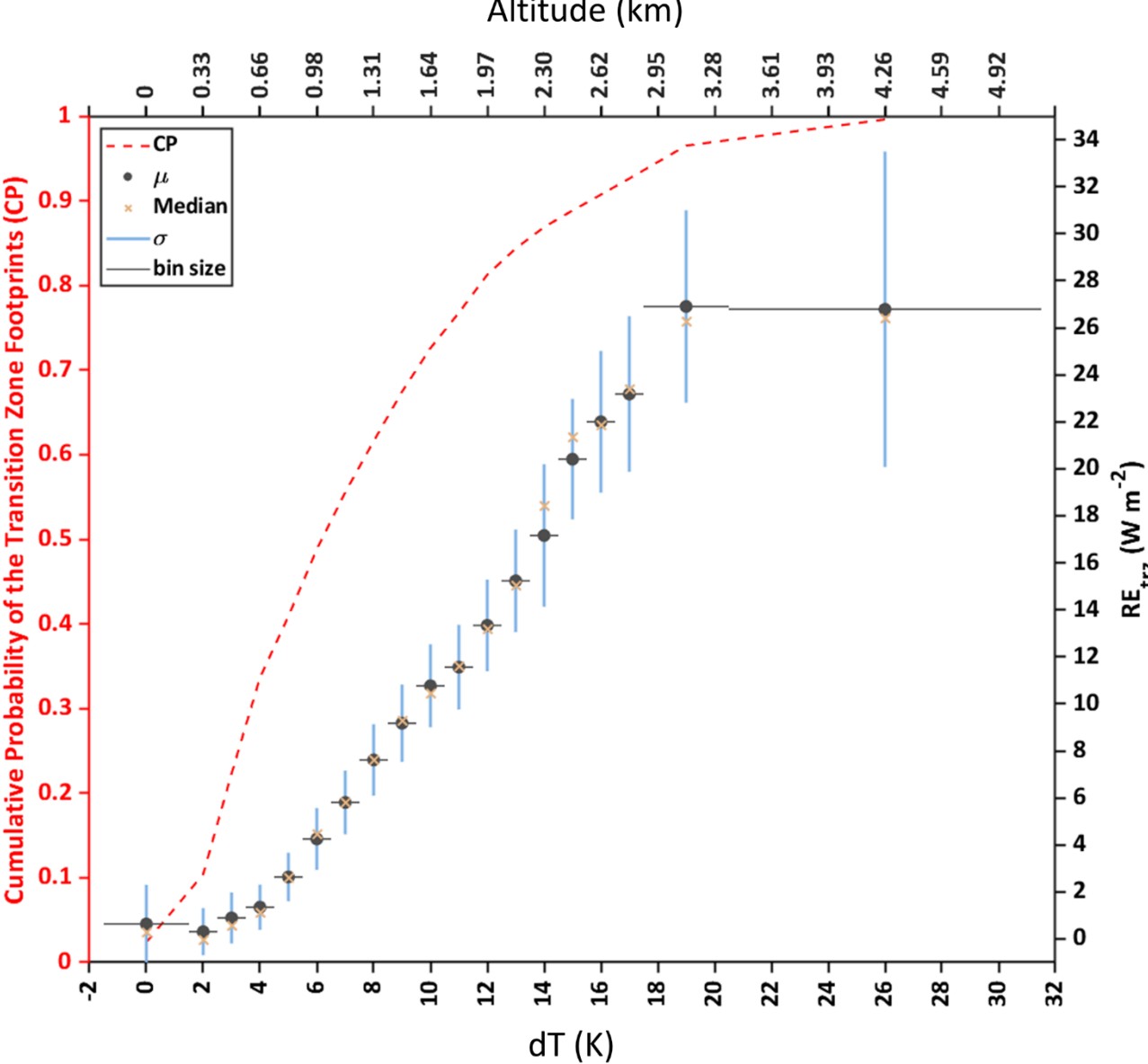

**Figure 5.** Cumulative probability (left axis) and $RE_{trz}$ (right) of the transition zone footprints analyzed as a function of dT. The vertical blue lines, black circles and yellow crosses indicate the standard deviation ($\sigma$), mean ($\mu$) and median of the $RE_{trz}$ values in each dT bin, respectively. The horizontal black lines show the width of each dT bin. The bottom X axis shows the center of each dT bin, and the top X axis shows the height associated with the dT value given in the bottom axis. The altitudes given in this figure are calculated following the estimates of the mean tropospheric temperature lapse rate for the study area (6.1 K/km) given in Mokhov and Akperov (2006).

## 4 Summary, Conclusions, and implications for atmospheric research

In the present study, a method for quantification of the broadband longwave radiative effects of the transition zone at TOA ($RE_{trz}$) during daytime over the ocean based on satellite observations and radiative transfer simulations was proposed. Specifically, $RE_{trz}$ was computed as the difference between the longwave irradiance as measured by CERES (Clouds and the Earth's Radiant Energy System) and the clear-sky irradiance as computed by Santa Barbara DISORT Atmospheric Radiative Transfer (SBDART) model run for the same place and moment, with the input data from ERA5 reanalysis. The identification of the transition zone conditions (CERES footprints) is based on MODIS (Moderate Resolution Imaging Spectroradiometer) products following the Schwarz et al. (2017) method, and 3783 CERES footprints have been found for the analyzed area in the Southeast Atlantic Ocean for August 2010. The uncertainty of the method for RE estimation was assessed by means of applying the same approach on clear-sky regions. This approach was applied to the data recorded by the CERES and MODIS sensors onboard Aqua platform during August 2010 over the South-East Atlantic Ocean. The results obtained from this analysis can be summarized as follows:

- The transition zone occurs over vast areas which makes it possible to observe its TOA radiative signature in radiative measurements at a spatial resolution as coarse as that of CERES.

- The methodology proposed in the present study is capable of quantifying the radiative effects of transition conditions with a wide range of characteristics with an accuracy of about $\pm 3.7$ W m$^{-2}$ at 95% confidence level, based on instantaneous satellite measurements and radiative transfer simulations.

- For the studied period and domain, $RE_{trz}$ is on average equal to 8.0 W m$^{-2}$ (heating effect; median: 5.4 W m$^{-2}$), although cases with $RE_{trz}$ with magnitudes as large as 50 W m$^{-2}$ were observed.

- Low-level transition zone conditions defined as those with suspension top height below 2 km (determined based on the difference between the layer top and surface temperature) on average produce a RE of about 4.6 W m$^{-2}$. The lowest layers (temperature difference less than 4 K) produce on average a RE of 0.8 Wm$^{-2}$.

- Although the overall effect of subpixel clouds on the calculated $RE_{trz}$ is likely to be small, we consider that our estimation of the $RE_{trz}$ should best be interpreted, cautiously, as an upper bound of the longwave radiative effects of the transition zone.

These results and those found by other studies show that the conditions corresponding to the transition zone have indeed an important  effect in the atmosphere, with a notable radiative signature in the longwave band, which deserves to be further investigated. The methodology presented in the current study provides the opportunity to gather information about the longwave radiative effects of homogeneous transition zone conditions with different characteristics. This information can be useful for characterizing the transition zone as an additional intermediate phase of particle suspension (class) between cloudy and cloud-free skies (containing aerosols or not) in the remote sensing algorithms, as well as in climatic, meteorological, and atmospheric studies. Nevertheless, this approach only provides information about the longwave radiative effects of the transition zone and the $RE_{trz}$ values given in the present study were obtained by analyzing only one month of data at a particular

study area. To be able to understand the role that the transition zone plays in the determination of the Earth's energy budget and the climate system, it is required to study the transition zone radiative effects in both longwave and shortwave spectral bands over larger domains and longer time spans. These aspects should be the matter of future research efforts.

## Appendix A

For all CERES footprints, we approximated the coordinates of the edges of CERES footprints assuming that they are
rectangularly shaped and then looked for MODIS pixels confined within the area scanned by CERES. To do so, we first
determined CERES viewing zenith angle ($\theta'$) from the CERES viewing zenith angle at surface ($\theta$) provided in the CERES
geolocation data according to Eq. A1:

$$\theta' = \sin^{-1}\left(\frac{R_e \sin(\theta)}{R_e + h_{sat}}\right) \qquad\qquad \text{Eq. A1}$$

where $R_e$ and $h_{sat}$ are the Earth radius (6371 km) and satellite altitude (705 km), respectively. To derive this equation, we
have assumed the Earth as a spherical object and applied the law of sines as illustrated in Figure A1 given below. Then, we
approximated the cross-scan length of the CERES footprints ($l_{cross\text{-}scan}$; km), according to Eq. A2, and assuming that Earth is
flat on the footprint scale and that CERES footprints are rectangularly shaped (see Figure A2 given below).

$$l_{cross-scan} = h_{sat}\left(tan(\theta' + 0.8127°) - tan(\theta' - 0.8127°)\right) \qquad\qquad \text{Eq. A2}$$

The along-scan length of the footprints ($l_{along\text{-}scan}$) was taken equal to 20 km (nadir resolution) as according to
https://ceres.larc.nasa.gov/instruments/ceres-operations/ the CERES instrument onboard Aqua spacecraft was operated in
cross-track mode for the study period. Afterwards, we integrated the processed MODIS data from 1-km resolution to CERES
native resolution by looking for MODIS pixels confined within the area scanned by CERES, considering equal weights for
all MODIS pixels.

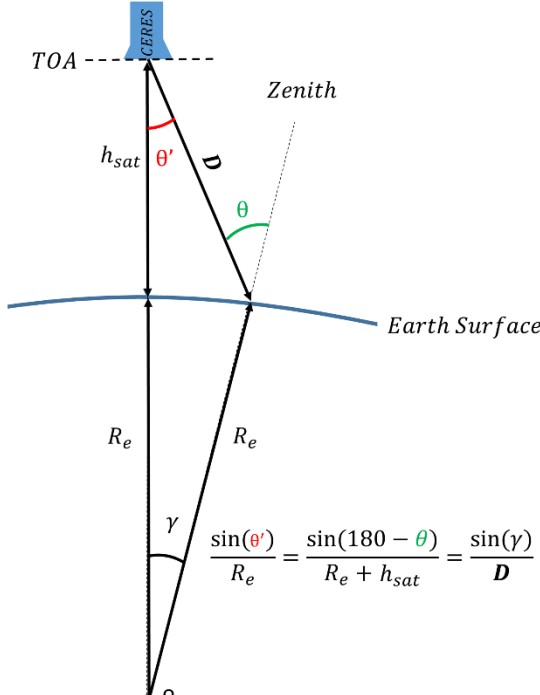

**Figure A1. Geometry for Eq. A1.**

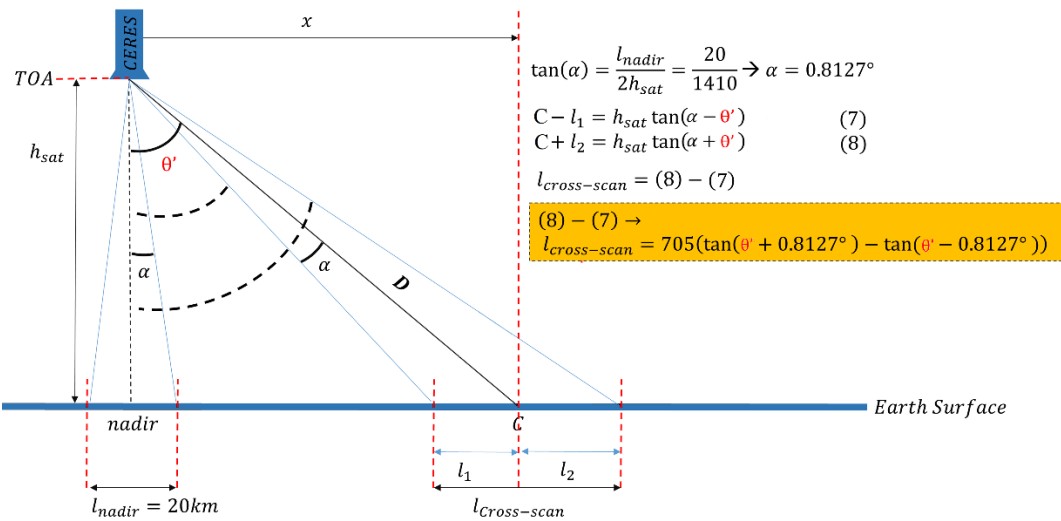

**Figure A2. Geometry for Eq. A2. Point C in this figure indicates the longitude at the center of the CERES footprint.**

**Code availability**

The source codes of Santa Barbra DISORT Atmospheric Radiative Transfer (SBDART) model can be accessed from https://github.com/paulricchiazzi/SBDART.git, last access: May 2021.

**Data Availability Statement**

All the data used in this study are publicly available and details on the datasets is provided in Section 2. The ERA5 reanalysis data used in this study are accessible from the Copernicus Climate Data Store at https://cds.climate.copernicus.eu/cdsapp#!/dataset/reanalysis-era5-pressure-levels?tab=overview and https://cds.climate.copernicus.eu/cdsapp#!/dataset/reanalysis-era5-single-levels?tab=overview, last access: February 2021. The MODIS data was obtained from the Level-1 and Atmosphere Archive & Distribution System (LAADS) Distributed Active Archive Center (DAAC) at https://ladsweb.modaps.eosdis.nasa.gov/, last access: February 2021. The CERES data was obtained from https://ceres-tool.larc.nasa.gov/ord-tool/products?CERESProducts=SSFlevel2_Ed4, last access: February 2021.

**Author contribution**

All authors contributed to the development of the initial idea, design of the study, interpretation of the results and reviewing the paper. Hendrik Andersen processed the MODIS data. Babak Jahani performed the computations, analyzed the data, and wrote the paper.

**Competing interests**

The authors declare that they have no conflict of interest.

**Acknowledgement**

This study is funded by the Spanish Ministry of Science and Innovation (project NUBESOL-2, PID2019-105901RB-I00) and Babak Jahani holds a FI-AGAUR PhD grant (2018FI_B_00830) provided by the Government of Catalonia (Universities and Research Secretariat) and the European Union.

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
