# Peer review of "Longwave Radiative Effect of the Cloud-Aerosol Transition Zone Based on CERES Observations"

_Atmospheric Chemistry and Physics, 2021_

## Author Comment (AC1)

Manuscript number: **#ACP-2021-421**

Title: **Longwave Radiative Effect of the Cloud-Aerosol Transition Zone Based on CERES Observations**

Submitted to: **Atmospheric Chemistry and Physics (ACP)**

Dear Reviewer,

On behalf of all authors, I would like to kindly thank you for your very constructive comments and suggestions, and as well for spending your precious time on reviewing our manuscript.

We are very pleased about the fact that the reviewer finds our manuscript interesting, sound and well written. All suggestions for minor refinements will be considered in the potential future submission of the manuscript. Please find below our answers to your major inquiries.

Note that the list of references mentioned in this document is given in the last page.

Sincerely yours,
Babak Jahani

**C#1.1)** It is important to clarify the sentence in Lines 90-91, which says "Indeed, for these pixels neither aerosol nor cloud optical property retrievals exist, yet they are classified as containing a cloud (Lost A), a non-cloud obstruction (Lost B), or were not processed at all in the cloud masking (Lost C)." Specifically, it should be clarified whether the study considers 1 km-size MODIS pixels as "cloudy" or "Lost" (most likely "Lost A") if the MYD06 cloud product does not include a positive retrieved value in the Scientific Data Set (SDS) "Cloud_Optical_Thickness", but includes a positive retrieved value in the SDS named "Cloud_Optical_Thickness_PCL". This occurs for partly cloudy 1 km-size pixels, in which clouds were detected for some, but not all 250 m-size subpixels. Clarifying this would be important because if such pixels were considered "Lost", CERES footprints containing many small clouds could be included in the transition zone statistics even if their total cloud fraction was well above 10% and their longwave effects came from cloud elements for which the MODIS cloud product did provide cloud property estimates.

**A#1.1)** Thank you for your constructive comment. For this analysis, we have followed the methodology presented in Schwarz et al. (2017) which is based on the variable labeled as "Cloud_Optical_Thickness". It is true that there may exist some conditions with developed scattered clouds in the subpixel scale of the MODIS pixels labeled as Lost A. It is also true that these small cloud fragments have an influence on the estimated radiative effects.
In addition, in line with this comment, **C#1.3** and **C#2.4** we have performed a statistical analysis, evaluating the links between the estimated $RE_{trz}$ values and the cloud fraction in the CERES transition zone footprints. The results of this analysis are presented below, in Figure 1.1. From the information provided in this figure, it can be seen that RE increases with cloud fraction, which confirms the abovementioned statement about the effect of clouds on the calculated $RE_{trz}$ values. However, the information provided in this figure also shows that for more than 75% of the CERES transition zone footprints that we have analyzed, the cloud fraction is below 5%. This implies that although subpixel clouds may have had an influence on our results given in Figure 3, their effect on the overall results of the analysis presented in section 6 should be rather small.
In the potential future submission, we will clearly state that we have performed our study based on the variable "Cloud_Optical_Thickness" and will add Figure 1.1 along with the discussion given above.

[Figure]

Figure 1.1. $RE_{trz}$ as a function of cloud fraction. Each cloud fraction bin given in this figure is 1% wide and the bar charts indicate the frequency of the transition zone footprints falling within the limits of each cloud fraction bin. The red circles and black vertical lines indicate the (bootstrapped) mean $RE_{trz}$ and the corresponding standard deviation for each cloud fraction bin, respectively.

- - - - - - - - - - - - - - - - - - - - - - - - - - - - - - - - - - - - - - - - - - - - - - - - - - - - - - - - - - - - - - - - - - - - - - - - - - - - - - - - - - -

**C#1.2)** Line 250 explains that the low-level transition zone effect of 0.8 W/m2 was calculated using the first four temperature difference (dT) bins in Fig. 4. However, it is not clear why four bins were used, rather than three, five, or more than five. This is a significant issue because Figure 4 suggests that the number of dT bins included into the low-level category can affect the results. It would help to explain why using the first four dT bins is a good choice. For example, could it be linked to a certain altitude range? It would also help to mention how the definition or the extent of the low-level category compares to the definition or extent in Eytan et al. (2020), which provided the radiative effect estimate of 0.75 W/m2 that was close to the 0.8 W/m2 in this paper.

**A#1.2)** In line 250, our aim was just to provide a quantitative comparison between our results and those found by Eytan et al. (2020). It should be noted that as you are aware (and mentioned in the manuscript), there are differences between our study and that of Eytan et al. (2020). One important difference is the fact that the study of Eytan et al. (2020) quantifies the radiative effect of the transition zone conditions occurring in the adjacency of the warm shallow cloud fields (defined as liquid-phase clouds with top temperatures warmer than 275 K). Whereas in our manuscript, we are studying the transition zone conditions with different characteristics occurring at different altitudes. Thus, for comparing our numbers with those of Eytan et al. (2020), we had to choose some of our studied transition zone conditions which could potentially match those analyzed in Eytan et al. (2020).
We have used the data corresponding to the CERES transition zone footprints falling within the

limit of the first four dT bins for this comparison because it covers the dT range (dT ≤ 3-4 K) which can be derived from Figure 2 given in Eytan et al. (2020). Indeed, the latter figure shows the cloud top brightness temperature and albedo along with the sea surface temperature for an example scene of what is defined as low cloud by them. This reason for selection of these four dT bins will be clearly mentioned in the potential future version of the manuscript. As the referee mentions, it is obvious that dT is connected with altitude, therefore, and also in line with a comment from the second referee, we will also modify Figure 4 by adding an additional X axis at the top to associate dT with altitude (please see A#2.4), and the term "low-level" used in other parts of the manuscript will be put in context. The altitudes given in this figure are calculated following the estimates of the mean tropospheric temperature lapse rate for the study area (6.1 K/km) given in Mokhov & Akperov (2006).

- - - - - - - - - - - - - - - - - - - - - - - - - - - - - - - - - - - - - - - - - - - - - - - - - - - - - - - - - -

C#1.3) The transition zone statistics include CERES footprints where up to 10% of MODIS pixels have neither aerosol nor cloud data. This criterion is very reasonable, but it allows including footprints where the cloud fraction can reach 10% (or much higher, depending on the treatment of partly cloudy MODIS pixels, as discussed in Point #1 above). Therefore, it could be interesting to discuss whether the transition zone radiative effect shows any statistical relationship to cloud fraction within the CERES footprint. This could be done either for all dT bins combined or for selected dT bins only.

A#1.3) Thank you for your constructive comment which is connected with your fist comment C#1.1. Therefore, please refer to our previous answer A#1.1. In particular, Figure 1.1. (above) corresponds to what the referee is asking: there is indeed a relationship between $RE_{trz}$ and the residual cloud fraction that may be present in the analyzed CERES footprints. As already mentioned, we will provide this information in the potential future submission of the manuscript. To clarify one point, however, statistics that we have provided correspond to the CERES transition zone footprints which consist of those with a "Lost" fraction (all lost classes together) greater than or equal to 90%, so there are only up to 10% of MODIS pixels which eventually have cloud data. In other words, in each analyzed CERES footprint at least 90% of MODIS pixels do not have cloud or aerosol data (they are "lost" pixels) while the contribution of all other classes combined ("Difficult", "Cloud", "Aerosol", "Clear") is less than or equal to 10% in these footprints. Therefore, the maximum cloud fraction in one of the analyzed CERES footprints might be 10% (and, as it can be seen in Figure 1.1 above, for more than 75% of the CERES transition zone footprints that we have analyzed, the cloud fraction is below 5%).

**References**

Eytan, E., Koren, I., Altaratz, O., Kostinski, A. B., & Ronen, A. (2020). Longwave radiative effect of the cloud twilight zone. *Nature Geoscience*, *13*(10), 669–673. https://doi.org/10.1038/s41561-020-0636-8

Mokhov, I. I., & Akperov, M. G. (2006). Tropospheric lapse rate and its relation to surface temperature from reanalysis data. *Izvestiya - Atmospheric and Ocean Physics*, *42*(4), 430–438. https://doi.org/10.1134/S0001433806040037

Schwarz, K., Cermak, J., Fuchs, J., & Andersen, H. (2017). Mapping the twilight zone-What we are missing between clouds and aerosols. *Remote Sensing*, *9*(6), 1–10. https://doi.org/10.3390/rs9060577

---

## Author Comment (AC2)

Manuscript number: **#ACP-2021-421**

Title: **Longwave Radiative Effect of the Cloud-Aerosol Transition Zone Based on CERES Observations**

Submitted to: **Atmospheric Chemistry and Physics (ACP)**

Dear Reviewer,

On behalf of all authors, I would like to kindly thank you for your very constructive comments and suggestions, and as well for spending your precious time on reviewing our manuscript.

We are very pleased about the fact that the reviewer finds our manuscript interesting, sound and well written. All suggestions for minor refinements will be considered in the potential future submission of the manuscript. Please find below our answers to your major inquiries.

Note that the list of references mentioned in this document is given in the last page.

Sincerely yours,
Babak Jahani

**C#2.1**) The "conventional" transition zone study focused on low level clouds such as Eytan et al. (2020) referenced in line 55. However, this paper does not mention if they looked at the transition zone near low level clouds. This is important because high thin cirrus could have similar LW effects but the cloud processes of thin cirrus clouds are completely different from low level clouds. This needs to be clarified.

**A#2.1**) As you mentioned the study of Eytan et al. (2020) is focused on the transition zone near low-level clouds, whereas our study involves, in principle, transition zone conditions at any given altitude. Specifically, in our study, we are proposing a method applicable for quantifying the longwave radiative effects of transition zone conditions with a wide range of characteristics present at various altitudes. This fact has been mentioned at different places in the manuscript and Figure 4 has been provided to prove the matter, as dT is connected with altitude.
In line with this comment and the comment **C#2.4** we will modify Figure 4 by adding an additional X axis at the top to associate dT with altitude (please see Figure 2.3 below, **A#2.4**).It can be seen that most transition zone situations (85%) correspond to suspensions with the top level at less than 2 km, and all of them, at less than 5 km, so still below the cirrus cloud levels.

- - - - - - - - - - - - - - - - - - - - - - - - - - - - - - - - - - - - - - - - - - - - - - - - - - - - - - - - - - - - - - - - - -

**C#2.2**) The classification of undefined pixels (Lost A, Lost B, Lost C) is useful. However, the paper lacks the description of how to match MODIS pixels to CERES footprints. Since this paper mainly presents a method to estimate the longwave effects of the transition zone, matching MODIS pixels to CERES footprints is a critical step, and it should be described.

**A#2.2**) Thank you for your constructive comment. Information about the matching of MODIS pixels to CERES footprints is given below and will be provided as an annex in the potential future submission of the manuscript (as we believe giving this information in the main text divert the reader from the main ideas of the research):

For all CERES footprints we approximated the coordinates of the edges of CERES footprints assuming that they are rectangularly shaped and then looked for MODIS pixels confined within the area scanned by CERES. To do so, we first determined CERES viewing zenith angle ($\theta'$) from the CERES viewing zenith angle at surface ($\theta$) provided in the CERES geolocation data according to Eq. 1:

$$\theta' = \sin^{-1}(\frac{R_e \sin (\theta)}{R_e+h_{sat}}) \hspace{3cm} \textit{Eq. 1}$$

where $R_e$ and $h_{sat}$ are the Earth radius (6371 km) and satellite altitude (705 km), respectively. To derive this equation, we have assumed the Earth as a spherical object and applied the law of sines as illustrated in Figure 2.1 given below. Then, we approximated the cross-scan length of the CERES footprints ($l_{cross-scan}$; km), according to Eq. 2, and assuming that Earth is flat on the footprint scale and that CERES footprints are rectangularly shaped (see Figure 2.2 given below).

$$l_{cross-scan} = h_{sat} \, (tan(\theta' + 0.8127°) - tan (\theta' - 0.8127°)) \hspace{2cm} \textit{Eq. 2}$$

The along-scan length of the footprints ($l_{along-scan}$) was taken equal to 20 km (nadir resolution) as according to https://ceres.larc.nasa.gov/instruments/ceres-operations/ the CERES instrument onboard Aqua spacecraft was operated in cross-track mode for the study period. Afterwards, we

integrated the processed MODIS data from 1-km resolution to CERES native resolution by looking for MODIS pixels confined within the area scanned by CERES, considering equal weights for all MODIS pixels.

[Figure]

**Figure 2.1**. Schematic description of reasoning behind Eq. 1.

[Figure]

**Figure 2.2.** Schematic description of the reasoning behind Eq. 2. Point C in this figure indicates the longitude at the center of the CERES footprint.

- - - - - - - - - - - - - - - - - - - - - - - - - - - - - - - - - - - - - - - - - - - - - - - - - - - - - - - - - - - - - - - - - -

**C#2.3)** CERES products provide both radiances and fluxes. The authors used LW radiance without no mentioning the reason not using the LW in the product. Is the sub-footprint cloud variability that makes the radiance-to-flux conversion difficult? Some discussions are necessary.

**A#2.3)** We have chosen to use radiances in our study rather than fluxes to provide a more direct comparison between the simulations and observations regardless of the sky condition.

Specifically, CERES instrument directly measures radiances, from which the irradiances (fluxes) are estimated using the empirical Angular Distribution Models (ADMs) explained in Loeb et al. (2005). Also, as the reviewer points out, estimation of irradiance from the radiance measured in a given direction requires accounting for the angular dependence in the radiance field, which is a strong function of the physical and optical characteristics of the scene (such as suspension fraction, optical depth and phase). For this reason, to apply the ADMs, it is important to have information about the optical and physical characteristics of the suspension within the CERES field of view and such information is not available for the transition zone conditions. According to your comment, reasoning for using radiances rather than fluxes will be provided in the potential future submission of the manuscript.

- - - - - - - - - - - - - - - - - - - - - - - - - - - - - - - - - - - - - - - - - - - - - - - - - - - - - - - - - - - - - - - -

C#2.4) The definition of temperature dT is not clear. It seems dT is the difference between surface air temperature and cloud top temperature (lines 227-235). It is hard for me to comprehend very small values of dT. What is the physical meaning when dT is very small? Is it because clouds are very low? Is it because of sub-pixel clouds in MODIS observations that makes cloud top temperature appears low? Some discussions are necessary.

A#2.4) Thank you for your valuable comment. According to your comment, additional (and clearer) discussion about how dT is computed and what it means will be provided in the potential future submission. Also, in line with this comment, we will add an additional horizontal axis to Figure 4. to link dT with altitude (please see Figure 2.3 given below).
Specifically, dT is the difference between the near surface air temperature and the suspension top temperature if the MODIS pixels labeled as "LOST A" were clouds. Let us further explain this: the MODIS pixels labeled as "LOST A" represent conditions that have been labeled as cloudy by MODIS cloud mask, whereas for them optical depth has not been retrieved (the algorithm has failed to retrieve). However, as they have been identified as cloud by the cloud mask, cloud top temperature has been retrieved for them.
As the reviewer says, the small values of dT correspond to transition zone conditions occurring very close to the sea surface. Indeed, according to Adebiyi et al. (2020), low-level clouds (cloud top height < 3 km) dominate the southeast Atlantic between July and October although mid-level clouds are as well relatively common over this region with cloud-top heights typically placed between 5 and 7 km. The information provided in Figure 2.3 (which will be the new version of current Figure 4 in the revised manuscript) also suggests that the majority of the transition zone conditions that we have studied are below 3 km.

[Figure]

**Figure 2.3.** Cumulative probability (left axis) and REtrz (right) of the transition zone footprints analyzed as a function of dT (bottom horizontal axis). The vertical blue lines, black circles and yellow crosses indicate the standard deviation (σ), mean (µ) and median of the REtrz values in each dT bin, respectively. The horizontal black lines show the width of each dT bin. The top horizontal axis indicates the altitude associated with each dT value given in the bottom horizontal axis. Note: the altitudes given in this figure have been computed according to the tropospheric temperature laps rate given in Mokhov & Akperov (2006) for the study region (6.1 K/km).

- - - - - - - - - - - - - - - - - - - - - - - - - - - - - - - - - - - - - - - - - - - - - - - - - - - - - - - - -

**C#2.5**) Line 34: "a phase called transition zone". "phase" has been used several times for the transition zone (e.g., line 40, line 217: "a phase of particles between the cloudy and socalled cloud-free skies..", line 289: "an important phase of particle suspensions..", line 293: "intermediate phase of particle suspension..".) To me the transition zone is not another phase of matters (e.g., solid, liquid, vapor). Even clouds contain liquid drops, ice crystals, and water vapor. I would use "a special region" to distinguish from clouds and cloud-free areas.

**A#2.5)** Thank you for your suggestion. We agree that the use of "phase" may be somewhat misleading, so we will avoid using it in the potential future submission.

- - - - - - - - - - - - - - - - - - - - - - - - - - - - - - - - - - - - - - - - - - - - - - - - - - - - - - -

**C#2.6)** Line 81: "homogenous" -> homogeneous

**A#2.6)** The corresponding text will be corrected accordingly.

- - - - - - - - - - - - - - - - - - - - - - - - - - - - - - - - - - - - - - - - - - - - - - - - - - - - - - -

**C#2.7)** Line 85: "These products were obtained for all MODIS-Aqua granules that contain data in the region 0° E – 15° E and 10° S –30° S during August 2010, which their data spreads over the area between 21° W – 21° E and 10° N –50° S." Not understand.

**A#2.7)** In this sentence we are trying to explain which granules (images) were used in our research. First, they corresponded to August 2010. Second, a granule was selected if it contained at least one data point (pixel) within the region 0° E – 15° E and 10° S –30° S. It turned out that the spatial extent of the data corresponding to the granules keeping these conditions covered an area between 21°W-21°E and 10°N-50°S.

- - - - - - - - - - - - - - - - - - - - - - - - - - - - - - - - - - - - - - - - - - - - - - - - - - - - - - -

**C#2.8)** Line 85: "MODIS-Aqua". I would use Aqua MODIS (e.g, Minnis 2011).

**A#2.8)** The corresponding text will be corrected accordingly.

- - - - - - - - - - - - - - - - - - - - - - - - - - - - - - - - - - - - - - - - - - - - - - - - - - - - - - -

**C#2.9)** Line 275: "3783 cases have been found…" I would change it to 3783 CERES footprints.

**A#2.9)** The corresponding text will be corrected accordingly.

- - - - - - - - - - - - - - - - - - - - - - - - - - - - - - - - - - - - - - - - - - - - - - - - - - - - - - -

**C#2.10)** Not sure if the boxplot inset of Figure 3 is necessary since all information is already available from the cumulative distribution and median and mean values indicated. If it do not provide additional information, it would be better to remove it.

**A#2.10)** Thank you for your suggestion. As you mentioned, the box plot provides a summary of the results given in the main figure. Although it does not provide new information, it may make it easier for the reader to comprehend the results faster. However, according to your comment, we will remove it in the potential future submission.

- - - - - - - - - - - - - - - - - - - - - - - - - - - - - - - - - - - - - - - - - - - - - - - - - - - - - - -

**C#2.11)** In the bar chart in Figure 3, should we expect to the sum of LOST A, B, and C to be one? I might have missed something. It would be nice to add some description in the caption so that the potential reader could see it immediately.

**A#2.11)** Thank you for your suggestion. In this bar chart, the sum of all classes ("Lost A", "Lost B", "Lost C", "Difficult", "Cloud", "Aerosol" and "Clear") equals one. According to your suggestion, we will

modify the caption and the legend of the figure in the potential future submission, to make it clearer that the white section of the bar chart involves the latter classes (difficult, cloud, aerosol, clear).

**References**

Adebiyi, A. A., Zuidema, P., Chang, I., Burton, S. P., & Cairns, B. (2020). Mid-level clouds are frequent above the southeast Atlantic stratocumulus clouds. *Atmospheric Chemistry and Physics*, *20*(18), 11025–11043. https://doi.org/10.5194/acp-20-11025-2020

Eytan, E., Koren, I., Altaratz, O., Kostinski, A. B., & Ronen, A. (2020). Longwave radiative effect of the cloud twilight zone. *Nature Geoscience*, *13*(10), 669–673. https://doi.org/10.1038/s41561-020-0636-8

Loeb, N. G., Kato, S., Loukachine, K., Manalo-Smith, N., & Doelling, D. R. (2005). Angular Distribution Models for Top-of-Atmosphere Radiative Flux Estimation from the Clouds and the Earth's Radiant Energy System Instrument on the Terra Satellite. Part I: Methodology. *Journal of Atmospheric and Oceanic Technology*, *22*, 338–351.

Mokhov, I. I., & Akperov, M. G. (2006). Tropospheric lapse rate and its relation to surface temperature from reanalysis data. *Izvestiya - Atmospheric and Ocean Physics*, *42*(4), 430–438. https://doi.org/10.1134/S0001433806040037

Schwarz, K., Cermak, J., Fuchs, J., & Andersen, H. (2017). Mapping the twilight zone-What we are missing between clouds and aerosols. *Remote Sensing*, *9*(6), 1–10. https://doi.org/10.3390/rs9060577

---

## Author Response (AR1)

Manuscript number: **#ACP-2021-421**

Title: **Longwave Radiative Effect of the Cloud-Aerosol Transition Zone Based on CERES Observations**

Submitted to: **Atmospheric Chemistry and Physics (ACP)**

Dear Reviewers,

On behalf of all authors, I would like to kindly thank you for your very constructive comments and suggestions, and as well for spending your precious time on reviewing our manuscript.

We are very pleased about the fact that the reviewers find our manuscript interesting, sound and well written. All suggestions for minor refinements have been considered in the new submission of the manuscript. Please find below our answers to your major inquiries.

Note that the list of references mentioned in this document is given in the last page.

Sincerely yours,
Babak Jahani

**C#1.1)** It is important to clarify the sentence in Lines 90-91, which says "Indeed, for these pixels neither aerosol nor cloud optical property retrievals exist, yet they are classified as containing a cloud (Lost A), a non-cloud obstruction (Lost B), or were not processed at all in the cloud masking (Lost C)." Specifically, it should be clarified whether the study considers 1 km-size MODIS pixels as "cloudy" or "Lost" (most likely "Lost A") if the MYD06 cloud product does not include a positive retrieved value in the Scientific Data Set (SDS) "Cloud_Optical_Thickness", but includes a positive retrieved value in the SDS named "Cloud_Optical_Thickness_PCL". This occurs for partly cloudy 1 km-size pixels, in which clouds were detected for some, but not all 250 m-size subpixels. Clarifying this would be important because if such pixels were considered "Lost", CERES footprints containing many small clouds could be included in the transition zone statistics even if their total cloud fraction was well above 10% and their longwave effects came from cloud elements for which the MODIS cloud product did provide cloud property estimates.

**A#1.1)** Thank you for your constructive comment. For this analysis, we have followed the methodology presented in Schwarz et al. (2017) which is based on the variable labeled as "Cloud_Optical_Thickness". It is true that there may exist some conditions with developed scattered clouds in the subpixel scale of the MODIS pixels labeled as Lost A. It is also true that these small cloud fragments have an influence on the estimated radiative effects.
In addition, in line with this comment, and comments **C#1.3** and **C#2.4**, we have performed a statistical analysis, evaluating the links between the estimated $RE_{trz}$ values and the cloud fraction in the CERES transition zone footprints. **We have added a new figure (Figure 4; Please see this figure at page # 12 in the clean manuscript) describing the results of this analysis in the revised manuscript.** From the information provided in this figure, it can be seen that $RE_{trz}$ increases with cloud fraction, which confirms the abovementioned statement about the effect of clouds on the calculated $RE_{trz}$ values. However, the information provided in this figure also shows that for more than 75% of the CERES transition zone footprints that we have analyzed, the cloud fraction is below 5%. This implies that although subpixel clouds may have had an influence on our results given in Figure 3, their effect on the overall results of our analysis should be rather small.
**In the revised manuscript, we have clearly mentioned that we have performed our study based on the variable "Cloud_Optical_Thickness" (please refer to line # 95 in the clean manuscript) and the abovementioned explanations are located at lines # 232 to 251 in the clean manuscript**.

- - - - - - - - - - - - - - - - - - - - - - - - - - - - - - - - - - - - - - - - - - - - - - - - - - - - - - - - - - - -

**C#1.2)** Line 250 explains that the low-level transition zone effect of 0.8 W/m2 was calculated using the first four temperature difference (dT) bins in Fig. 4. However, it is not clear why four bins were used, rather than three, five, or more than five. This is a significant issue because Figure 4 suggests that the number of dT bins included into the low-level category can affect the results. It would help to explain why using the first four dT bins is a good choice. For example, could it be linked to a certain altitude range? It would also help to mention how the definition or the extent of the low-level category compares to the definition or extent in Eytan et al. (2020), which provided the radiative effect estimate of 0.75 W/m2 that was close to the 0.8 W/m2 in this paper.

**A#1.2**) In (former) line 250, our aim was just to provide a quantitative comparison between our results and those found by Eytan et al. (2020). It should be noted that as you are aware (and mentioned in the manuscript), there are differences between our study and that of Eytan et al. (2020). One important difference is the fact that the study of Eytan et al. (2020) quantifies the radiative effect of the transition zone conditions occurring in the adjacency of the warm shallow cloud fields (defined as liquid-phase clouds with top temperatures warmer than 275 K). Whereas in our manuscript, we are studying the transition zone conditions with different characteristics occurring at different altitudes. Thus, for comparing our numbers with those of Eytan et al. (2020), we had to choose some of our studied transition zone conditions which could potentially match those analyzed in Eytan et al. (2020).

We have used the data corresponding to the CERES transition zone footprints falling within the limit of the first four dT bins for this comparison because it covers the dT range (dT $\leq$ 3-4 K) which can be derived from Figure 2 given in Eytan et al. (2020). Indeed, the latter figure shows the cloud top brightness temperature and albedo along with the sea surface temperature for an example scene of what is defined as low cloud by them.

**The reason for selection of these four dT bins as well as description on some of the existing differences between our study and that of (Eytan et al., 2020) has been given at lines 292 to 303 in the revised manuscript.**

As the referee mentions, it is obvious that dT is connected with altitude, therefore, in line with this comment and also a comment from the second referee (please see **A#2.4**), we have modified Figure 4 (**presented as Figure 5 in the revised manuscript; please see Figure 5 at page # 15 in the clean manuscript**) by adding an additional X axis at the top to associate dT with altitude according to the estimates of the mean tropospheric temperature lapse rate for the study area (6.1 K/km) given in Mokhov & Akperov (2006). In addition, we have put the term "low-level" in context by defining it as conditions with suspension top altitude less than 2 km. **Please see the lines # 259 to 291 of the clean manuscript.**

- - - - - - - - - - - - - - - - - - - - - - - - - - - - - - - - - - - - - - - - - - - - - - - - - - - - - - - - - - - - - -

**C#1.3**) The transition zone statistics include CERES footprints where up to 10% of MODIS pixels have neither aerosol nor cloud data. This criterion is very reasonable, but it allows including footprints where the cloud fraction can reach 10% (or much higher, depending on the treatment of partly cloudy MODIS pixels, as discussed in Point #1 above). Therefore, it could be interesting to discuss whether the transition zone radiative effect shows any statistical relationship to cloud fraction within the CERES footprint. This could be done either for all dT bins combined or for selected dT bins only.

**A#1.3**) Thank you for your constructive comment which is connected with your fist comment (**C#1.1**). Therefore, please refer to our previous answer **A#1.1**. **In particular, Figure 4 in the revised manuscript corresponds to what the referee is asking**: there is indeed a relationship between RE$_{trz}$ and the residual cloud fraction that may be present in the analyzed CERES footprints. As already mentioned, we have provided this information in the revised manuscript.

To clarify one point, however, statistics that we have provided correspond to the CERES transition zone footprints which consist of those with a "Lost" fraction (all lost classes together) greater than or equal to 90%, so there are only up to 10% of MODIS pixels which eventually have

cloud data. In other words, in each analyzed CERES footprint at least 90% of MODIS pixels do not have cloud or aerosol data (they are "lost" pixels) while the contribution of all other classes combined ("Difficult", "Cloud", "Aerosol", "Clear") is less than or equal to 10% in these footprints. Therefore, the maximum cloud fraction in one of the analyzed CERES footprints might be 10% (and, as it can be seen in **Figure 4**, for more than 75% of the CERES transition zone footprints that we have analyzed, the cloud fraction is below 5%). In addition, the information provided in the bar chart in Figure 3 also shows that although the criteria explained in section 2.1 for the selection of transition zone footprints allows up to 10% of contamination by "non-lost" classes, the fraction of these classes combined in the transition zone footprints analyzed is on average about 5%. **Please see lines # 190 to 195 and 232 to 251 of the clean manuscript**.

- - - - - - - - - - - - - - - - - - - - - - - - - - - - - - - - - - - - - - - - - - - - - - - - - - - - - - - - - - - - - -

**C#1.4**) Line 18: I suggest adding "the" between "onboard" and "Aqua".
Line 28: I suggest replacing "regardless of" by "without considering".
Line 39: I suggest adding "the" between "that" and "transition".
Line 81: I suggest adding "the" between "for" and "identification".

**A#1.4**) Thank you for your suggestions. The corresponding text has been modified according to your suggestions.

- - - - - - - - - - - - - - - - - - - - - - - - - - - - - - - - - - - - - - - - - - - - - - - - - - - - - - - - - - - - - -

**C#1.5**) Lines 83-84: It would help to clarify whether the 3 km or 10 km resolution MYD04 aerosol product was used.

**A#1.5**) Thank you for your comment. We have used the 10km resolution aerosol product (MYD04_L2). **The name of the product has been clearly mentioned at line #87 in the revised manuscript**.

- - - - - - - - - - - - - - - - - - - - - - - - - - - - - - - - - - - - - - - - - - - - - - - - - - - - - - - - - - - - - -

**C#1.6**) Lines 96-97: I suggest rewording "number of ocean MODIS pixels more than or equal to 75% of the expected ≈400 pixels falling within CERES field of view (FOV)" to something like "the number of MODIS ocean pixels equals or exceeds 75% of the ≈400 pixels expected to fall within the CERES field of view (FOV)".

**A#1.6**) Thank you for your suggestion. The corresponding sentence has been modified according to your suggestion. **Please see lines # 102 to 104 of the revised manuscript**.

**C#2.1**) The "conventional" transition zone study focused on low level clouds such as Eytan et al. (2020) referenced in line 55. However, this paper does not mention if they looked at the transition zone near low level clouds. This is important because high thin cirrus could have similar LW effects but the cloud processes of thin cirrus clouds are completely different from low level clouds. This needs to be clarified.

**A#2.1**) As you mentioned the study of Eytan et al. (2020) is focused on the transition zone near low-level clouds, whereas our study involves, in principle, transition zone conditions at any given altitude. Specifically, in our study, we are proposing a method applicable for quantifying the longwave radiative effects of transition zone conditions with a wide range of characteristics present at various altitudes. This fact has been mentioned at different places in the manuscript and Figure 4 (**labeled as Figure 5 in the revised manuscript; please see this figure at page # 15 in the clean manuscript**) has been provided to prove the matter.
In line with this comment as well as the comments **C#2.4** and **C#1.2** we have modified Figure 4 (**presented as Figure 5 in the revised manuscript**) by adding an additional X axis at the top to associate dT with altitude according to the estimates of the mean tropospheric temperature lapse rate for the study area (6.1 K/km) given in Mokhov & Akperov (2006). It can be seen that most transition zone situations (85%) correspond to suspensions with the top level at less than 2 km, and all of them, at less than 5 km, so still below the cirrus cloud levels. **Please see the lines # 275 to 303 of the clean manuscript**.

- - - - - - - - - - - - - - - - - - - - - - - - - - - - - - - - - - - - - - - - - - - - - - - - - - - - - - - - - - - - -

**C#2.2**) The classification of undefined pixels (Lost A, Lost B, Lost C) is useful. However, the paper lacks the description of how to match MODIS pixels to CERES footprints. Since this paper mainly presents a method to estimate the longwave effects of the transition zone, matching MODIS pixels to CERES footprints is a critical step, and it should be described.

**A#2.2**) Thank you for your constructive comment. Information about the matching of MODIS pixels to CERES footprints is given below and has been provided in "**Appendix A**" in the revised manuscript (as we believe giving this information in the main text would divert the reader from the main ideas of the research).

- - - - - - - - - - - - - - - - - - - - - - - - - - - - - - - - - - - - - - - - - - - - - - - - - - - - - - - - - - - - -

**C#2.3**) CERES products provide both radiances and fluxes. The authors used LW radiance without no mentioning the reason not using the LW in the product. Is the sub-footprint cloud variability that makes the radiance-to-flux conversion difficult? Some discussions are necessary.

**A#2.3**) We have chosen to use radiances in our study rather than fluxes to provide a more direct comparison between the simulations and observations regardless of the sky condition. Specifically, CERES instrument directly measures radiances, from which the irradiances (fluxes) are estimated using the empirical Angular Distribution Models (ADMs) explained in Loeb et al. (2005). Also, as the reviewer points out, estimation of irradiance from the radiance measured in a given direction requires accounting for the angular dependence in the radiance field, which is a strong function of the physical and optical characteristics of the scene (such as suspension fraction, optical depth and phase). For this reason, to apply the ADMs, it is important to have

information about the optical and physical characteristics of the suspension within the CERES field of view and such information is not available for the transition zone conditions. According to your comment, reasoning for using radiances rather than fluxes has been provided in the revised manuscript. **Please see lines # 70 to 82 of the clean manuscript.**

- - - - - - - - - - - - - - - - - - - - - - - - - - - - - - - - - - - - - - - - - - - - - - - - - - - - - - - - - - - - - - - - - - - -

**C#2.4**) The definition of temperature dT is not clear. It seems dT is the difference between surface air temperature and cloud top temperature (lines 227-235). It is hard for me to comprehend very small values of dT. What is the physical meaning when dT is very small? Is it because clouds are very low? Is it because of sub-pixel clouds in MODIS observations that makes cloud top temperature appears low? Some discussions are necessary.

**A#2.4**) Thank you for your valuable comment. According to your comment, additional (and clearer) discussion about how dT is computed and what it means has been provided in the revised manuscript (**please refer to lines # 259 to 274**). Also, in line with this comment, and comments **C#2.1** and **C#1.2**, we have added an additional horizontal axis to former Figure 4 (**labeled as Figure 5 in the revised manuscript**) to link dT with altitude. **Please see Figure 5 at page # 15**. Specifically, dT is the difference between the near surface air temperature and the suspension top temperature if the MODIS pixels labeled as "LOST A" were clouds. Let us further explain this: the MODIS pixels labeled as "LOST A" represent conditions that have been labeled as cloudy by MODIS cloud mask, whereas for them optical depth has not been retrieved (the algorithm has failed to retrieve). However, as they have been identified as cloud by the cloud mask, cloud top temperature has been retrieved for them.

As the reviewer says, the small values of dT correspond to transition zone conditions occurring very close to the sea surface. Indeed, according to Adebiyi et al. (2020), low-level clouds (cloud top height < 3 km) dominate the southeast Atlantic between July and October although mid-level clouds are as well relatively common over this region with cloud-top heights typically placed between 5 and 7 km. The information provided in Figure 5 given in the revised manuscript also suggests that the majority of the transition zone conditions that we have studied are below 3 km. **Please see lines # 275 to 303 of the clean manuscript.**

- - - - - - - - - - - - - - - - - - - - - - - - - - - - - - - - - - - - - - - - - - - - - - - - - - - - - - - - - - - - - - - - - - - -

**C#2.5**) Line 34: "a phase called transition zone". "phase" has been used several times for the transition zone (e.g., line 40, line 217: "a phase of particles between the cloudy and socalled cloud-free skies..", line 289: "an important phase of particle suspensions..", line 293: "intermediate phase of particle suspension..".) To me the transition zone is not another phase of matters (e.g., solid, liquid, vapor). Even clouds contain liquid drops, ice crystals, and water vapor. I would use "a special region" to distinguish from clouds and cloud-free areas.

**A#2.5**) Thank you for your suggestion. We agree that the use of "phase" may be somewhat misleading, so as suggested by the referee, we have used a combination of the words "**region**" and "**condition**" (depending on the concept) in the revised manuscript. **Please see the trackchanges manuscript**.

- - - - - - - - - - - - - - - - - - - - - - - - - - - - - - - - - - - - - - - - - - - - - - - - - - - - - - - - - - - - - - - - - - - -

**C#2.6)** Line 81: "homogenous" -> homogeneous

**A#2.6)** The corresponding text has been **corrected** accordingly.

- - - - - - - - - - - - - - - - - - - - - - - - - - - - - - - - - - - - - - - - - - - - - - - - - - - - - - - - - - - - - - -

**C#2.7)** Line 85: "These products were obtained for all MODIS-Aqua granules that contain data in the region 0° E – 15° E and 10° S –30° S during August 2010, which their data spreads over the area between 21° W – 21° E and 10° N –50° S." Not understand.

**A#2.7)** In this sentence we are trying to explain which granules (images) were used in our research. First, they corresponded to August 2010. Second, a granule was selected if it contained at least one data point (pixel) within the region 0° E – 15° E and 10° S –30° S. It turned out that the spatial extent of the data corresponding to the granules keeping these conditions covered an area between 21°W-21°E and 10°N-50°S. **This sentence has been modified in the revised manuscript (Please see lines # 88 to 91 of the clean manuscript).**

- - - - - - - - - - - - - - - - - - - - - - - - - - - - - - - - - - - - - - - - - - - - - - - - - - - - - - - - - - - - - - -

**C#2.8)** Line 85: "MODIS-Aqua". I would use Aqua MODIS (e.g, Minnis 2011).

**A#2.8)** The corresponding text has been **corrected** accordingly.

- - - - - - - - - - - - - - - - - - - - - - - - - - - - - - - - - - - - - - - - - - - - - - - - - - - - - - - - - - - - - - -

**C#2.9)** Line 275: "3783 cases have been found…" I would change it to 3783 CERES footprints.

**A#2.9)** The corresponding text has been **corrected** accordingly.

- - - - - - - - - - - - - - - - - - - - - - - - - - - - - - - - - - - - - - - - - - - - - - - - - - - - - - - - - - - - - - -

**C#2.10)** Not sure if the boxplot inset of Figure 3 is necessary since all information is already available from the cumulative distribution and median and mean values indicated. If it do not provide additional information, it would be better to remove it.

**A#2.10)** Thank you for your suggestion. According to your comment, we have removed it from Figure 3. **Please see Figure 3 at page # 10 in the clean manuscript**.

- - - - - - - - - - - - - - - - - - - - - - - - - - - - - - - - - - - - - - - - - - - - - - - - - - - - - - - - - - - - - - -

**C#2.11)** In the bar chart in Figure 3, should we expect to the sum of LOST A, B, and C to be one? I might have missed something. It would be nice to add some description in the caption so that the potential reader could see it immediately.

**A#2.11)** Thank you for your suggestion. In this bar chart, the sum of all classes ("Lost A", "Lost B", "Lost C", "Difficult", "Cloud", "Aerosol" and "Clear") equals one. According to your suggestion, we have modified the caption and the legend of the figure in the potential future submission, to make it clearer that the white section of the bar chart involves the latter classes (difficult, cloud, aerosol, clear). **Please see Figure 3 at page # 10 in the clean manuscript**.

**References**

Adebiyi, A. A., Zuidema, P., Chang, I., Burton, S. P., & Cairns, B. (2020). Mid-level clouds are frequent above the southeast Atlantic stratocumulus clouds. *Atmospheric Chemistry and Physics*, *20*(18), 11025–11043. https://doi.org/10.5194/acp-20-11025-2020

Eytan, E., Koren, I., Altaratz, O., Kostinski, A. B., & Ronen, A. (2020). Longwave radiative effect of the cloud twilight zone. *Nature Geoscience*, *13*(10), 669–673. https://doi.org/10.1038/s41561-020-0636-8

Loeb, N. G., Kato, S., Loukachine, K., Manalo-Smith, N., & Doelling, D. R. (2005). Angular Distribution Models for Top-of-Atmosphere Radiative Flux Estimation from the Clouds and the Earth's Radiant Energy System Instrument on the Terra Satellite. Part I: Methodology. *Journal of Atmospheric and Oceanic Technology*, *22*, 338–351.

Mokhov, I. I., & Akperov, M. G. (2006). Tropospheric lapse rate and its relation to surface temperature from reanalysis data. *Izvestiya - Atmospheric and Ocean Physics*, *42*(4), 430–438. https://doi.org/10.1134/S0001433806040037

Schwarz, K., Cermak, J., Fuchs, J., & Andersen, H. (2017). Mapping the twilight zone-What we are missing between clouds and aerosols. *Remote Sensing*, *9*(6), 1–10. https://doi.org/10.3390/rs9060577